# Dynamics of Ex Vivo Mesenchymal Stromal Cell Potency under Continuous Perfusion

**DOI:** 10.3390/ijms24119602

**Published:** 2023-05-31

**Authors:** Aneesha Doshi, Patrick Erickson, Matthew Teryek, Biju Parekkadan

**Affiliations:** Department of Biomedical Engineering, Rutgers University, Piscataway, NJ 08854, USA; aneesha.doshi@rutgers.edu (A.D.);

**Keywords:** bioreactor, MSC, ex vivo therapy, immunomodulation, immunotherapy, secretome, licensing

## Abstract

Mesenchymal stromal cells (MSCs) are a candidate for cell immunotherapy due to potent immunomodulatory activity found in their secretome. Though studies on their secreted substances have been reported, the time dynamics of MSC potency remain unclear. Herein, we report on the dynamics of MSC secretome potency in an ex vivo hollow fiber bioreactor using a continuous perfusion cell culture system that fractionated MSC-secreted factors over time. Time-resolved fractions of MSC-conditioned media were evaluated for potency by incubation with activated immune cells. Three studies were designed to characterize MSC potency under: (1) basal conditions, (2) in situ activation, and (3) pre-licensing. Results indicate that the MSC secretome is most potent in suppressing lymphocyte proliferation during the first 24 h and is further stabilized when MSCs are prelicensed with a cocktail of pro-inflammatory cytokines, IFNγ, TNFα, and IL-1β. The evaluation of temporal cell potency using this integrated bioreactor system can be useful in informing strategies to maximize MSC potency, minimize side effects, and allow greater control for the duration of ex vivo administration approaches.

## 1. Introduction

The field of immunotherapy has expanded to include cellular therapy, particularly in the fields of oncology and autoimmune diseases. Along with common cancer treatments such as radiation and chemotherapy, mesenchymal stromal cells (MSCs) are a candidate for immunotherapy due, in large part, to their secretome containing immunomodulatory factors and extracellular vesicles [1]. Immunoassays have demonstrated the capability of MSCs to suppress the activation of immune cells [2,3,4,5,6]. However, MSC potency is dynamic and their immunomodulatory behavior is largely dependent on conditions of their microenvironment, particularly the degree of inflammation present, with certain factors causing a switch in MSC-mediated immunomodulation [7,8]. Additionally, prelicensing, or exposing MSCs to a cocktail of inflammatory cytokines including IFNγ, TNFα, and IL-1β prior to their use as a therapy, has been found to promote an anti-inflammatory profile that is then maintained in further potency assays [9,10,11]. Previous research has shown that IFNγ alone is unable to trigger MSC suppression of T cell proliferation; however, in combination with TNFα or IL-1β, MSCs are activated, and their inhibitory affects are enhanced [12]. Despite their in vitro success, there has been limited clinical translation of intravascular MSC therapies due to issues of cell persistence and clot induction in vivo [13,14,15,16,17,18,19].

Ex vivo therapy has been developed as an alternative approach to control the delivery of MSC-secreted factors continuously into the blood stream. Investigators have studied concentrated conditioned medium from MSCs as a cell-free formulation [20,21], yet the safety and efficacy of such an approach has not been tested in human patients to date. Another approach to ex vivo therapy consists of MSCs seeded into a hollow fiber bioreactor with a semipermeable membrane and defined pore size, allowing for the interaction between the MSC-secreted factors and peripheral blood, assuring stable exposure to MSCs and mitigating the in vivo complications [22,23,24]. In the development of the system, it was found that MSCs adhered to the hollow fibers while maintaining their fibroid morphology as well as remaining viable under flow conditions [24]. Ex vivo MSC use has also led to the discovery of the successful reprogramming of blood cells after exposure to these seeded MSCs, as well as the dynamic and responsive nature of various factors in the MSC secretome, including metabolism and secreted cytokines [23]. First-in-human evaluation of ex vivo MSC therapy demonstrated regulation of the peripheral immune response in patients with systemic inflammation [22]. Continued research to understand when and under what conditions the MSCs are the most potent and deliver greater immunosuppressive factors can further ex vivo MSC therapy applications.

Despite these studies, there has been limited investigation into the temporal dynamics of MSC potency and how it changes based on the conditions of the microenvironment. To explore the dynamic potency of ex vivo MSCs, we leveraged a continuous perfusion system design that fractionates media in specific time intervals to model the existing therapy and allow us to characterize MSC behavior in greater depth [23,25]. Culture medium perfused through MSC-seeded bioreactors and was collected in outlet fractions at 2 h intervals and were used in subsequent assays to quantify their potency using a T cell stimulation bioassay and a cytokine analysis. Once basal MSC potency was established, we introduced pro-inflammatory cytokines into the system to understand how the potency changed in response to an inflammatory mediator in real time. Finally, we prelicensed the MSCs by exposing them to a cocktail of proinflammatory cytokines including IFNγ, TNFα, and IL-1β before they were seeded to the bioreactor to determine whether the prelicensing would activate and enhance MSC potency and if this potency would be maintained throughout the system. Building upon previous students conducted by Allen et al. [23], these studies allowed us to quantify and understand the potency of MSCs at extremely small intervals and elucidate the dynamic nature of the potency. We discovered that in a basal state, MSCs were most potent during the first 24 h, and introducing pro-inflammatory cytokines promoted a shift towards an immunosuppressive profile in the MSC secretions. Prelicensing enhanced the potency significantly, causing the greatest suppression of immune cells, throughout the perfusion. These results allow us to understand the behavior and responses of MSCs in this model to ultimately create a potent therapeutic device, an MSC-seeded bioreactor, to deliver a more powerful and controlled therapeutic over the duration of treatment.

## 2. Results

### 2.1. Baseline Viability, Potency, and Secreted Factor Release of Immobilized MSCs in the Continuous Perfusion System

The first study was conducted to understand the behavior of MSCs in their basal state in a bioreactor perfusion system. The perfusion was conducted for 72 h to quantify their behavior over time. To study MSC exosome production, the culture medium used was an exosome-depleted alpha minimum essential medium (α-MEM). Minimization of xenogeneic signaling through exosomes, in theory, can reduce artifacts when studying an MSC secretome. Exosomes were removed from fetal bovine serum (FBS) to make the α-MEM via a recommended protocol of ultracentrifugation for 18 h at 100,000× *g* [26]. Studies conducted on MSCs harvested from the system after perfusion confirmed that the exosome-free serum had no impact on the viability of the MSCs (Appendix A). Exosomes were depleted for future characterization of secreted–exosome testing at the molecular level.

MSCs were seeded in semi-permeable hollow fiber bioreactors. Bioreactors were seeded with 1 × 10^6^ (1 M) and 3 × 10^6^ (3 M) MSCs per device (*n* = 3) and then perfused at a constant rate of 0.5 mL/h. The pore size of 0.2 μm allowed for the MSCs to remain within the bioreactor while the medium continuously flowed through. As the medium interacted with the seeded MSCs, it collected MSC-secreted factors and cytokines before it flowed out to be collected into small fractions (Figure 1A). Fractions of perfused medium with the MSC supernatant were collected every 2 h after perfusion began. Every 24 h after perfusion began, completed fractions were collected and stored at 4 °C until assayed at the end of perfusion to determine their individual potencies. The downstream studies are further depicted in Figure 1B. The dynamic baseline LDH (U/L) secretion by MSCs during 72 h continuous perfusion was quantified from each fraction by a Cedex Bio Analyzer. Fractions were cocultured with CD3/CD28–stimulated and CFSE–stained PBMCs for 72 h before being analyzed via flow cytometry for T cell proliferation. Finally, cytokine production was measured from the supernatant of the potency assay.

First, a viability assay was conducted using a Cedex Bio Analyzer to measure the amount of lactate dehydrogenase (LDH) (U/L) produced by both 1 M and 3 M MSCs throughout the perfusion. LDH is an intracellular enzyme that is released by necrotic cells through the plasma membrane [27]. LDH secretions were quantified from each fraction throughout the duration of the perfusion to detect MSC cytotoxicity and determine whether MSCs are able to maintain viability under perfusion conditions. For both the cell doses, there was an initially high value of LDH, which quickly decreased by 10 h and remained at a relatively stable value for the remainder of the 72 h perfusion (Figure 2A). This suggests that there may be some cell death when introduced to the system within the first 24 h, but for the remainder of the perfusion the MSCs are able to remain viable. Harvesting of MSCs from the bioreactor after 24 h of perfusion yielded fewer cells than were input, and at lower viabilities (Appendix A), although the harvesting process itself may confound results as it may not be completely efficient for full cell removal or without injury to cells. 

To determine the dynamic MSC potency, each fraction was cocultured for 2 days with CD3/CD28 stimulated PBMCs to determine their effect on PBMC. A nonstimulated static control was used to create the proliferative gates. This control consisted of untreated perfusion medium, pure exosome-depleted ɑ-MEM that had been perfused through an acellular control bioreactor, a bioreactor seeded with 0 M MSCs, cocultured with stimulated PBMCs, indicated as 0 M. From these gates, the percent proliferation of each fraction was determined. The gating strategy and representative histograms can be seen in Appendix A. A fold change in proliferation was determined by finding the difference between the fraction concentration and the control concentration, then dividing by the control concentration. These fold changes for each fraction are plotted (Figure 2B). In the baseline state, there appears to be a shift in MSC potency after about 24 h of perfusion for the 1 M cell dose. Before 14 h, the fractions suppress lymphocyte proliferation up to 29.7%, but after 21 h, the 1 M cell dose fractions begin promoting lymphocyte proliferation up to 15.3%. The fractions from the 3 M cell dose appear to be suppressive throughout all the fractions from the 72 h. To further analyze the baseline potencies, statistical analyses were conducted to understand the differences between dosage groups and fractions. A mixed-effects ANOVA and one-way repeated measures ANOVA with Tukey’s multiple comparisons were conducted to find statistically significant differences between the two cell dosages and the acellular control (1 M, 3 M, 0 M), and between the fractions of these dosages, to determine which fractions were significantly different. There were significant differences found between the fractions of the 3 M dosage and the 0 M dosage at different timepoints as well as specific times between the 1 M and 3 M dosages.

Results from the one-way repeated measures ANOVA with Tukey’s multiple comparisons conducted between the different dosages at larger time periods was also performed (Table 1). For the first 24 h of perfusion, there was the greatest significance between the 0 M and 3 M cell groups, and no significance between the 1 M and 3 M cell groups, in suppressing lymphocyte proliferation. For the rest of the perfusion, there were statistically significant differences between the 1 M cell groups and the other two dosages, likely due to the 1 M fractions promoting proliferation after 24 h. For the 0 M and 3 M cell group, there is decreasing significance and no significance in suppressing proliferation at the end of perfusion.

The supernatants from the proliferation assays were assessed to determine the production of cytokines in each coculture well (Figure 3) [23,28]. A negative control of fractions of medium perfused through bioreactors seeded with 0 M MSCs was included to understand the secreted factors from the PBMCs solely and determine the effect of each MSC fraction on the PBMCs in the well. Measurement of IL-6 shows an initial peak in concentration which decreases until 24 h before stabilizing. IFNγ concentrations from each well vary depending on timepoint and increase after 24 h in the 1 M dosage fractions. TNFα measurements remain below the control and slightly increase after 24 h. Similarly, with TGFα, concentrations remain consistently below the control value. To further analyze the baseline cytokine release differences, statistical analyses were conducted to understand the differences between dosage groups and fractions. A mixed-effects ANOVA and one-way repeated measures ANOVA with Tukey’s multiple comparisons were conducted to find statistically significant differences between the three cell dosages (0 M, 1 M, 3 M) and between the fractions of these dosages to determine which concentrations were significantly different. Results from all the cytokines tested can be found in Appendix A. A table of the *p*-values represented on the figures can be found in Appendix A. These results give insight into the secretions from stimulated PBMCs when cocultured in perfusion medium from MSCs in their basal state.

### 2.2. Modeling Inflammation: Viability, Potency, and Secreted Factor Release of MSCs Introduced to Proinflammatory Cytokine Medium

With an understanding of the baseline MSC potency and behavior in our dynamic perfusion system, we next studied how MSC potency would change under inflammatory conditions. To mimic inflammation, exosome-depleted α-MEM was treated with a pro-inflammatory cytokine cocktail: 10 ng/mL of IFNγ, 10 ng/mL of TNFα, and 10 ng/mL of IL-1β. Concentrations used were consistent with those in previous in vitro coculture assays cited [9,29,30]. From the results of the first perfusion, a dosage of 3 M cells was chosen to be tested in triplicate to determine if the potency of MSCs could be increased throughout the duration of the perfusion. Bioreactors were seeded with 3 × 10^6^ (3 M) MSCs per device (*n* = 3) and then perfused at 0.5 mL/h. For the first 24 h of perfusion, bioreactors with 3 M cells (*n* = 3) were perfused with untreated exosome-depleted α-MEM. At 24 h, we began perfusing the treated cytokine medium throughout the system for the remaining 48 h of perfusion. The remainder of the perfusion and analyses performed remained consistent with the first study.

The viability assay (Figure 4A) shows a similar initial peak in LDH production by the cells, which steadily decreases until 20 h, where it remains stable until about 60 h. After this point, there is a steady increase in LDH production until the end of perfusion, indicating a greater release of LDH by the MSCs at the end of the perfusion.

Using the lymphocyte activation assay with fractions of MSC-secreted factors, we observed a difference in MSC potency after 24 h (Figure 4B). For the first 24 h, the 3 M cell dosage prevented lymphocyte proliferation as before and started to wane closer to 24 h. When the cytokine medium was introduced as a step change, there appeared to be a renewal of potency, as the amount of lymphocyte suppression increased after 24 h, as opposed to the first study, where the potency decreased after this time. The suppression slightly decreased but remained consistently below the control, suggesting an increase in potency because of the introduction of cytokine medium. From a multiple unpaired student’s *t*-test conducted on the two cell dosage groups, 3 M vs. 0 M, more fractions in this study compared to the baseline showed statistical significance in suppressing lymphocyte proliferation than basal medium conditions. A table of the *p*-values represented on the figure can be found in Appendix A. This suggests that the fractions collected after the introduction of proinflammatory medium were significant in suppressing lymphocyte proliferation compared to the negative control. 

From the cytokine analysis, there is a clear difference in fractions from before the cytokine medium and the fractions after the cytokine medium was introduced, at 24 h. The concentrations of the pro-inflammatory TNFα, IL-1β, and IFNγ have been included to show the clear difference in fractions before and after the introduction of inflammatory medium (Appendix A). Since these agents were spiked into the medium, they served to verify the successful spiking and delivery to MSCs, though any interpretation of these cytokines would be confounded, as the medium was collected from the MSC and PBMC coculture. Cytokines with notable dynamic changes have been included in Figure 5. To further analyze differences in cytokine concentration, statistical analyses were conducted to understand the differences between dosage groups and fractions. A multiple unpaired student’s *t*-test was conducted on the two cell dosage groups (0 M and 3 M) to find statistically significant differences between the cytokine levels. A table of *p*-values can be found in Appendix A. IL-6 concentrations in cocultures from the first 24 h fractions looked similar to the previous perfusion, although the fold increase in concentration was greater. There was an initial peak and then a decrease until the 36 h fraction. However, in later fractions, there was a sharp increase in concentrations, similar to those of the earlier fractions. For IL-1Ra, there was a strong correlation between the introduction of the cytokine medium, particularly IL-1β, and the decrease in IL-1RA concentrations in the later fractions after 24 h, as IL-1Ra blocks the activity of IL-1β. These cytokine dynamics suggest that there is a correlation between the introduction of pro-inflammatory TNFα, IL-1β, and IFNγ and the resulting secreted factors by the MSCs in perfusion as well as the PBMCs in the potency assay. Notably, our previous studies found that the mean residence time of the tubing connecting the medium source to the bioreactor was 5.3 h, and the step change should have fully arrived by 6.7 h [31]. The dramatic changes in cytokines occur roughly 6–8 h following the step change, indicating the MSCs likely began to respond immediately to the change in their inflammatory environment.

This study indicated that MSCs in our dynamic perfusion system were sensitive to changes in the microenvironment, such as proinflammatory cytokines, and dynamically shifted their potency and secreted factors in response to inflammation. As there was a correlation between the proinflammatory cytokines and an increase in MSC potency compared with the baseline, we can understand further how the MSCs would function in delivering this ex vivo therapeutic to a patient with inflammation.

### 2.3. Prelicensing: Viability, Potency, and Secreted Factor Release of Prelicensed MSCs 

As our previous study indicated, MSCs exposed to proinflammatory cytokines experience enhanced potency, promoting further suppression of immune cells, likely due to a behavior switch into an anti-inflammatory state [1]. Previous research has indicated that prelicensing, or exposing MSCs to a combination of inflammatory cytokines prior to their use in cell therapy, promotes an anti-inflammatory profile that is then maintained in further potency assays [9,10,11]. Additionally, the combination of IFNγ with TNFα or IL-1β is required to induce MSC activation and prelicensing [12]. This induction of MSC-mediated immunosuppression enhances their benefit as a cellular therapeutic. Therefore, this study was designed to understand whether MSC prelicensing with a combination of IFNγ, TNFα, of IL-1β would enhance their potency, and, if so, to determine whether this potency would be translated to our system and would be maintained for the duration of perfusion.

To conduct this study, exosome-depleted α-MEM was treated with an identical pro-inflammatory cytokine cocktail containing 10 ng/mL of IFNγ, 10 ng/mL of TNFα, and 10 ng/mL of IL-1β [9,29,30]. MSCs were plated in a T-75 flask and cultured for 24 h in this cocktail medium. After 24 h of exposure to this cocktail medium, MSCs were trypsinized and directly seeded to the bioreactors (*n* = 3), with 3 M cells per bioreactor. The perfusion conditions were identical to the first study.

The viability assay reveals an LDH secretion profile very similar to the baseline study (Figure 6A). There is a peak in concentration within the first 3 h of perfusion, but, by 9 h, the concentration stabilizes and remains consistent throughout the remainder of perfusion. A similar profile to the baseline state indicates that the highest release of LDH occurs within the first 12 h of perfusion, and that the cells remain viable and stable throughout the rest of the perfusion.

A lymphocyte potency assay was again conducted with the collected fractions (Figure 6B). The potency from prelicensed MSC fractions compared with the control was greater than all other conditions in this study. Each fraction from the duration of the 72 h perfusion was now significantly suppressing lymphocyte proliferation. A table of *p*-values can be found in Appendix A. Additionally, the potency profile appears to be the most stable of the conditions in this study, with all fractions being in a closer range of suppression compared with those of the previous studies.

Figure 7 presents the dynamic secreted factor profiles from the stimulated PBMCs cocultured with fractions from the prelicensed perfusion. To further analyze differences in cytokine concentration, statistical analyses were conducted to understand the differences between dosage groups and fractions. A multiple unpaired student’s *t*-test was conducted on the two cell dosage groups (0 M and 3 M) to find statistically significant differences between the cytokine levels. A table of *p*-values can be found in Appendix A. For TNFα, there is an overall increase in concentration compared WITH the control as the fraction time increases. Around the 27 h fraction, there is a general shift where the concentration begins to be greater than the control, which is maintained in the later fractions except for a few, such as the 35 h, 37 h, and 65 h fractions. A similar increase in IL-1β as fraction time increases is present, with the greatest increase happening between the 21–25 h fractions. As with the baseline study, the concentration of IL-6 is highest in the earliest fraction cocultures and decreases with time before stabilizing around the 13 h fraction. However, the stable concentrations are at a much higher fold than the control compared with the baseline perfusion. Finally, looking at the IL-1Ra profile, there is a similar profile as the baseline study as well, with a negative fold change ranging from −0.431 to −0.816 but remaining between those values. Results from all the cytokines tested are in Appendix A.

To compare dynamic MSC potency of the three studies, the three studies were compared to each other in Figure 8. Prelicensing the MSCs enhances their potency significantly and to the greatest degree compared with the other studies.

## 3. Discussion

Here, we describe the study of dynamic MSC potency and secreted factors using an ex vivo bioreactor and a continuous perfusion system. MSCs are a potent candidate for cellular immunotherapy due to their secretion of soluble factors and their dynamic responses to factors in their microenvironments [32,33]. This study was conducted to further characterize MSC potency and behavior in this hollow-fiber bioreactor system via a continuous perfusion system. This model of attached MSCs exposed to continuous flow through a porous membrane can be relevant for engrafted cells after transplantation that are near vascular/interstitial flow environments or as a direct comparison to ex vivo MSC bioreactor therapy. These studies focused on paracrine effects of MSCs on immune cells, though future studies can explore other targets of MSC potency such as vascular and epithelial tissues [34,35].

In the first study, we established a baseline profile for the dynamic MSC viability, potency, and secreted factors in an unstimulated state. Cell dosages were chosen based on previous research to determine a dose-dependent effect of MSC potency [22,23]. While higher cell dosages are used in clinical applications of the MSC-seeded hollow-fiber bioreactors, these dosages allow a small-scale study of our system while still being able to quantify MSC behavior in a continuous perfusion system. A perfusion time of 72 h was chosen to quantify the dynamic secretions of MSCs over a longer duration and allow them to become accustomed to the conditions of the study before introducing factors to influence their behavior, such as inflammatory cytokines. This amount of time also represents the maximum use of a clinical MSC device per the ratings of existing hemofilters in human patients. We were able to see a dynamic response at 24, 48, and 72 h in the various perfusions. We established that the MSCs were viable throughout the duration of perfusion (Figure 2A). In the potency assay, around the 24 h fraction, with the lack of inflammation in the microenvironment, we observed an MSC behavior switch from an anti-inflammatory to a pro-inflammatory profile, consistent with previous literature [7] (Figure 2B). The 1 M cell dose remained significantly different from the control dose of 0 M, while the 3 M cell dose and the 0 M cell dose were not statistically different (Table 1). After analyzing the secreted factors from the potency assay, we could see a peak in IL-6 concentrations in the earliest fraction, and a decrease in concentration before a stabilization as the fraction number increased. We could also see the dynamic differences in IFNγ, TNFα, and TGFα, giving insight into the baseline factors from the coculture (Figure 3).

From here, we created a model of inflammation by introducing a pro-inflammatory cytokine cocktail to a 3 M cell dose and perfusing this medium for 48 h. In Figure 4A, there is a similar decrease in LDH secretion over time with an increase towards the end of perfusion, indicating that the cells may be losing viability after longer-term exposure to these cytokines. Results from the potency assay suggest that the introduction of pro-inflammatory cytokines can induce and sustain an immunosuppressive profile by the MSCs (Figure 4B). At the 24 h fraction, with the introduction of the cytokine medium, the MSCs do not switch behavior, but are able to continue suppressing lymphocyte proliferation. From the secreted factors, there is a shift in concentrations at and after the 24 h fraction, suggesting that the presence of pro-inflammatory cocktail medium affects the secretion of other cytokines (Figure 5). 

Finally, we prelicensed a cell dose of 3 M MSCs before using them in our system. Prelicensing, or activating, MSCs switches their behavior to an anti-inflammatory state before they are used in a cell therapy product, enabling them to be in the most potent state from the start of the therapy. The viability assay indicates that there is an initial high concentration of LDH, but that quickly stabilizes for the remainder of perfusion to a value similar to that of the first viability study, indicating that the prelicensed cells are viable throughout the perfusion (Figure 6A). The potency assay suggests that this form of activating the MSCs has the greatest effect on suppression of lymphocytes when cultured with MSC secretions (Figure 6B). The magnitude of lymphocyte suppression was greatest in this assay compared with the previous two potency assays, and potency was maintained for the entirety of the perfusion, with all the fractions having significant differences in potency compared with the negative control. This is further corroborated by Figure 8, comparing the three potency assays and showing the magnitude of suppression that prelicensing MSCs provides. From the secreted factors in Figure 7 there was an increase in concentrations of TNFα and IL-1β across the fractions, particularly around the 24 h fractions. Additionally, there is a similar peak in IL-6 in the earliest fractions which stabilizes later in the fractions, although at a higher fold than that of the first perfusion.

These results have implications to the use of MSCs in vivo in terms of dose and duration of MSC therapy. For dosage, several approximations are used to put this study in the context of in vivo use cases. In this report, a dose of MSCs at 3 M was found to have more stable potency than 1 M over several days. By scaling the number of MSCs per control volume of the bioreactor (~1 mL), this equates to a blood concentration of 1 M/mL or 3 M/mL in the respective doses tested. Several reports have evaluated an intravenous dose of MSCs at 200 M and, if administered to an adult with circulating blood volume of ~5 L, this would amount to 0.04 M/mL at a theoretical peak concentration, or about 10–100 times lower than what was studied in our report. For ex vivo MSC bioreactor administration use, the dose used in research bioreactors scales linearly to the volume of the device itself and equates to 250 M or 750 M cells per bioreactor, with a nominal volume of 250 mL for clinical hemofilters. Further exploration of dosage and scaling is warranted, though, based on these order of magnitude analyses, it would seem that MSC administration by intravenous routes may fall below the necessary cell concentrations in the blood to induce potency, whereas MSC bioreactor administration can control for secretome dosing at these concentrated levels. Additionally, ex vivo MSC bioreactor administration allows for the mitigation of common adverse events with intravenous administration, such as thromboembolism and fibrosis [13,14,15,16,17,18,19]. 

In a transplant setting (either local or systemic), it is likely that MSCs will encounter an inflammatory environment with elevated levels of cytokines such as TNFα, IL-1β, and/or IFNγ. These conditions likely favor a model were basal MSCs become activated in situ and thus have a revived immunopotent secretome after 24 h, as we observed in Figure 5. The pharmacodynamic effects of MSCs would likely follow two phases (or ~two days) of lymphocyte suppression assuming that MSC viability is maintained through the transplant process. In the case of ex vivo MSC bioreactor use, this administration route further assures MSC viability during this period of time and suggests that a duration of use from 24–48 h would provide stable immunotherapeutic effects. We have not explored individual cytokine stimulation effects on MSCs nor their suppression capacity after 72 h, so this restricts further view on long-term immune changes.

Overall, these studies show that our system is able to deliver the benefits of MSC immunotherapy in an ex vivo manner, mitigating the in vitro complications while allowing for more control over the dose and duration of the therapy. Consistent with previous literature, prelicensing the cells led to the most potent immunosuppressive profile of secreted factors from the MSCs, providing insight into how to deliver a more effective therapy to patients using our system. With a deeper understanding of temporal MSC potency and the ability to increase the potency of our therapeutic, we are able to further the use of MSCs as a powerful ex vivo cell therapy product for autoimmune disease and inflammation.

## 4. Materials and Methods

### 4.1. Human Cell Cultures

Deidentified human mesenchymal stromal cells were isolated from bone marrow aspirate obtained from Lonza Bioscience (Catalog #: 1 M-125, Lonza, Walkersville, MD, USA) with donor consent. Rutgers University did not require the study to be reviewed by an ethics committee because the commercial vendor, Lonza Bioscience, has their own ethical consent process for obtaining samples. All MSC lines were obtained from a single female donor and cryopreserved at passage number 2. MSC cell lines used in the perfusion systems were expanded and used at passage number 3. Cells used in this study displayed multipotent differentiation to adipose, cartilage, and bone cell types and had an immunophenotype of CD105+, CD90+, CD73+, CD34−, and CD45−. 

### 4.2. Media Preparation

The medium used for MSC-seeding and perfusions consisted of exosome-depleted α-minimum essential medium (α-MEM), containing 10% fetal bovine serum (FBS), 1% antibiotic-antimycotic, and 0.2% gentamycin for a duration of 72 h. Exosomes were isolated from a 10% HyClone FBS (Fisher Scientific, Roskilde, Denmark), via ultracentrifugation at 100,000× *g* (L8-70 M, Beckman, Brea, CA, USA) for 18 h and collecting the resulting exosome-free supernatant [26]. The medium used to culture PBMCs was comprised of RPMI 1640 medium, consisting of 1% antibiotic-antimycotic. 

### 4.3. Bioreactor MSC Seeding

The bioreactors (Spectrum Laboratories, Dominguez, CA, USA; C02-P20U-05) contain an internal volume of 1.5 mL and a surface area of 28 cm^2^. The nine 0.5 mm diameter hollow fibers within the bioreactors are comprised of polyethersulfone and are semi-permeable, with a pore size of 0.2 μm. MSCs were thawed from cryopreservation and counted via a NucleoCounter (NC-202, ChemoMetec, Lillerød, Denmark). For this smaller scale study to characterize the behavior of the seeded MSCs, the seeding densities were determined based on a prior study that used a seeding density of 3 M MSCs per bioreactor [23]. 1 M was chosen as a seeding density for the first study to determine the difference in potency between the two cell dosages. For the second and third study, the preferred dosage of 3 M was chosen. 

The desired cell number (1 × 10^6^ (1 M) or 3 × 10^6^ (3 M)) of live MSCs was suspended into 9 mL of exosome-depleted α-MEM and seeded onto PBS-primed bioreactor fibers through the extraluminal space. As excess medium flowed around the cells, the cells remained adhered to the hollow fibers of the bioreactors. The bioreactors were incubated at 37 °C to allow for cells to attach and remain seeded to the hollow fibers. They were then immediately used in the perfusion assay. A close-up representation of the MSC-seeded bioreactor fibers can be seen Figure 1A, and further studies using these MSC-seeded bioreactors can be found [22,23,24].

### 4.4. Perfusion and Fraction Collection

Bioreactors were perfused and fractions collected using methods described previously [25,31]. Briefly, culture medium for the perfusion was loaded into syringes and placed in a multi-channel syringe pump (New Era Pump Systems, NE-1600, Farmingdale, NY, USA) kept outside the incubator. Medium was dispensed at a constant rate of 0.5 mL/h through 1 m lengths of silicone tubing (Cole-Parmer, EW-96410-14, Vernon Hills, IL, USA) connecting each syringe to one of the bioreactors kept inside the incubator through an open port. A second 1 m length of silicone tubing connected the outlet of each bioreactor to a fraction collector (BioFrac Fraction Collector, Bio-Rad 7410002, Hercules, CA, USA) outside the incubator with a custom modification allowing it to collect from multiple bioreactors simultaneously. Medium was collected into each fraction for two hours. For experiments requiring a step-change in media composition, each bioreactor was supplied by two syringes containing the different media and dispensed by different syringe pumps. These two syringes dispensed into two ports of a four-way stopcock which was used to control which source flowed out the third port and into the silicone tube leading to the bioreactor.

Every 24 h, the collected fractions were stored at 4 °C until the end of the 72 h perfusion. Immediately after the 72 h, the potency assay was set up using the fractions. The fractions were then immediately stored at −80 °C.

### 4.5. Cytokine Activation

Exosome-depleted α-MEM was treated with a pro-inflammatory cytokine cocktail. Previous assays have reported quantifiable changes in MSC efficacy and potency after prelicensing with a concentration of 10 ng/mL of IFNγ, 5–15 ng/mL of TNFα, and 10 ng/nL of IL-1β [11,23,36]. Final concentrations of 10 ng/mL of IFNγ, 10 ng/mL of TNFα, and 10 ng/mL of IL-1β (R&D Systems, Minneapolis, MN, USA) were chosen for consistency with similar MSC potency studies. The medium was prepared directly before perfusion and placed in syringes. Bioreactors were seeded with 3 × 10^6^ (3 M) MSCs per device (*n* = 3) and then perfused at 0.5 mL/h. For the first 24 h of perfusion, bioreactors with 3 M cells (*n* = 3) were perfused with untreated exosome-depleted a-MEM. At 24 h, we switched the valve to change the input medium to the cytokine-treated medium, perfusing the treated cytokine medium throughout the system for the remaining 48 h of perfusion.

### 4.6. Prelicensing

The 3 M MSCs were seeded in a T175 flask (*n* = 3 flasks) for 24 h in exosome-depleted α-MEM to allow for the MSCs to adhere and adjust to the culture conditions. After 24 h, the medium in the flask was replaced with exosome-depleted α-MEM treated with an identical cytokine cocktail as the second perfusion containing 10 ng/mL of IFNγ, 10 ng/mL of TNFα, and 10 ng/mL of IL-1β (R&D Systems, Minneapolis, MN, USA). This cocktail medium was created just before being added to the flask. After 24 h of culture in this cocktail medium, MSCs were trypsinized from each flask and directly seeded to the bioreactors (*n* = 3), with 3 M cells per bioreactor.

### 4.7. Measurement of LDH Production

Fractions of perfused medium containing MSC secretions were collected from each perfusion and analyzed using a Cedex Bio Analyzer (Roche Diagnostics, Basel, Switzerland), according to manufacturer’s instructions, to determine the enzymatic activity of Lactate Dehydrogenase (LDH) released by MSCs throughout the perfusion. Concentration (U/L) was detected via a LDH Bio Test kit for the Cedex Bio Analyzer (Roche Diagnostics, Basel, Switzerland).

### 4.8. PBMC Isolation

Peripheral blood mononuclear cells (PBMCs) were obtained from the New York Blood Center (New York City, NY, USA) and freshly isolated from leukopaks via density gradient centrifugation using Ficoll-Paque (GE Healthcare). PBMCs from a single donor were used throughout the studies in the paper to reduce donor variability. Informed consent was obtained for each donor according to vendor-specific protocols and IRB review. Human samples were obtained by commercial vendors for research purposes with donor consent.

### 4.9. CFSE Labeling for PBMC Proliferation

PBMCs were thawed from cryopreservation and resuspended in phosphate-buffered saline (PBS). Carboxyfluorescein succinimidyl ester (CFSE) stock (5 mM in DMSO; Biotin, CA, USA) stored at −20 °C was thawed and incubated with PBMCs in PBS at 1 μL/mL for five minutes at room temperature without light. Cells were washed and resuspended in RPMI culture medium and cultured overnight. A sample of PBMCs were left unstained as a control for flow cytometry.

### 4.10. PBMC Stimulation

Human CD3/CD28 T Cell Activator (ImmunoCult™, Stem Cell Technologies, Vancouver, BC, Canada) was added to CFSE-stained PBMCs in culture medium at a concentration of 25 μL/mL. PBMCs were then added to a 96-well plate for high-throughput proliferation assay. After 24 h, fractions from perfusions were added to respective wells, for a final concentration of 150,000 cells/150 μL medium, or 1 × 10^6^ cells/mL. An image of this assay setup is shown in Figure 2A.

### 4.11. Flow Cytometric Analysis

After 72 h of incubation, supernatant from the coculture of PBMCs and MSC fractions was collected for multiplex analysis. PBMCs were collected and resuspended in MACS buffer. Proliferation was analyzed via flow cytometry on a FACSCanto II (BD Biosciences, Franklin Lakes, NJ, USA). Viable cells were gated based on forward/side scatter via BD FACSDiva, and lymphocyte populations were selected. Flow cytometry analysis was performed in FlowJo (FlowJo LCC, Ashland, OR, USA; version 8.7.3). Proliferative cell gates were set using a non-stimulated static control (no stimulation with CD3/CD28). A fold change in proliferation was determined by finding the difference between the fraction concentration and the control concentration, then dividing by the control concentration.

### 4.12. Analysis of Cytokine Production

The mixed medium supernatant from the potency assay consisting of 75 μL RPMI and 75 μL fraction medium was collected and frozen at −80 °C until assayed using a Human 10-Plex Cytokine Assay with Luminex xMAP technology (TNFα, IL-1β, IFNγ, IL-6, IL-10, IL-1Ra, TGFα).

### 4.13. Statistical Analysis

Statistical analyses were performed utilizing GraphPad Prism 9.0 (MacIntosh version). Statistical significance for potency differences and cytokine concentration differences between the normalized control and fraction was determined by mixed-effects and one-way repeated measures ANOVA with Tukey’s multiple comparisons post-test and Student’s *t*-test. Statistical significance was indicated by the following: * = significance at 0.05 level, ** = significance at 0.01 level, *** = significance at 0.001 level, **** = significance at 0.0001 level.

## Figures and Tables

**Figure 1 ijms-24-09602-f001:**
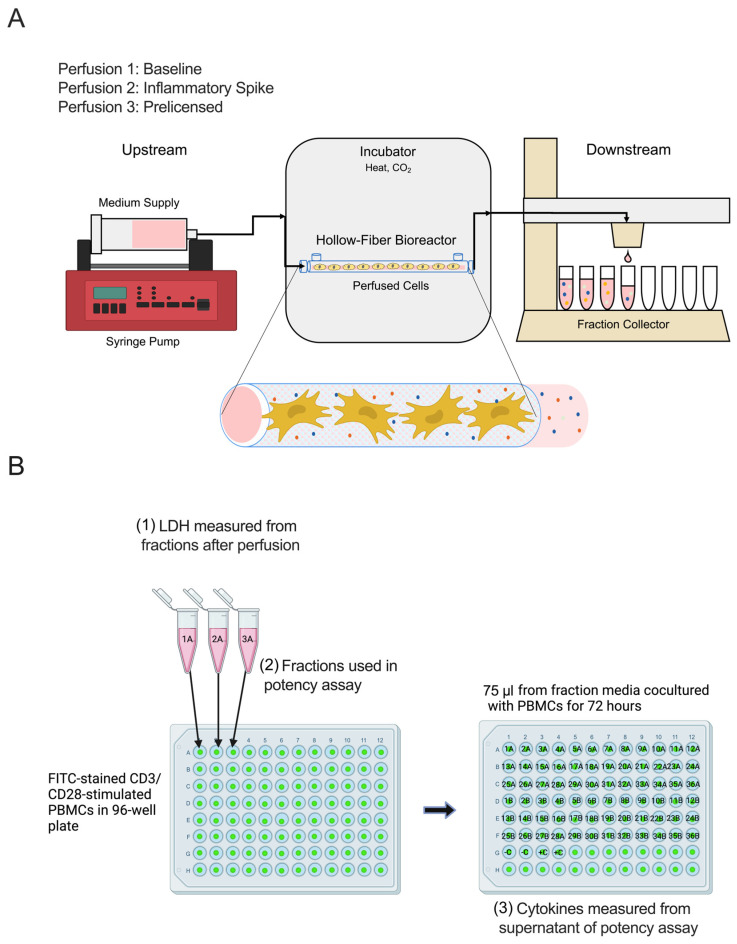
Study overview. (**A**) Schematic of hollow-fiber bioreactor seeded with MSCs under continuous perfusion. Bioreactors were seeded with 1 × 10^6^ (1 M) and 3 × 10^6^ (3 M) viable MSCs, incubated for 2 h at 37 °C, allowing the MSCs to adhere to the hollow fibers. Exosome–depleted α-MEM containing 10% FBS, 1% antibiotic-antimycotic, and 0.2% gentamycin was pumped continuously via a syringe pump through tubing into the MSC–seeded hollow-fiber bioreactors. As the medium perfused through the cells, it collected the secreted factors from the MSCs and was pumped to the downstream portion to be collected. Fractions of the MSC secretome were collected every 2 h for the duration of the 72 h perfusion. (**B**) Downstream studies conducted with fractions after perfusion. (1) Dynamic baseline LDH (U/L) secretion by MSCs during 72 h continuous perfusion was quantified from each fraction by Cedex Bio Analyzer. (2) Fractions were cocultured with CD3/CD28–stimulated and CFSE–stained PBMCs for 72 h before being analyzed via flow cytometry for T cell proliferation. For each well, the number indicates the time of each fraction, with fraction 1 being medium collected in the first 2 h of perfusion, fraction 2 being the next 2 h, etc., until the full 72 h of perfusion was completed. The letter (A or B) indicates the bioreactor the fraction was collected from (*n* = 3), while C indicates the positive and negative controls of PBMCs with/without CD3/CD28 stimulation and CFSE staining. (3) Finally, cytokine production was measured from the supernatant of the potency assay.

**Figure 2 ijms-24-09602-f002:**
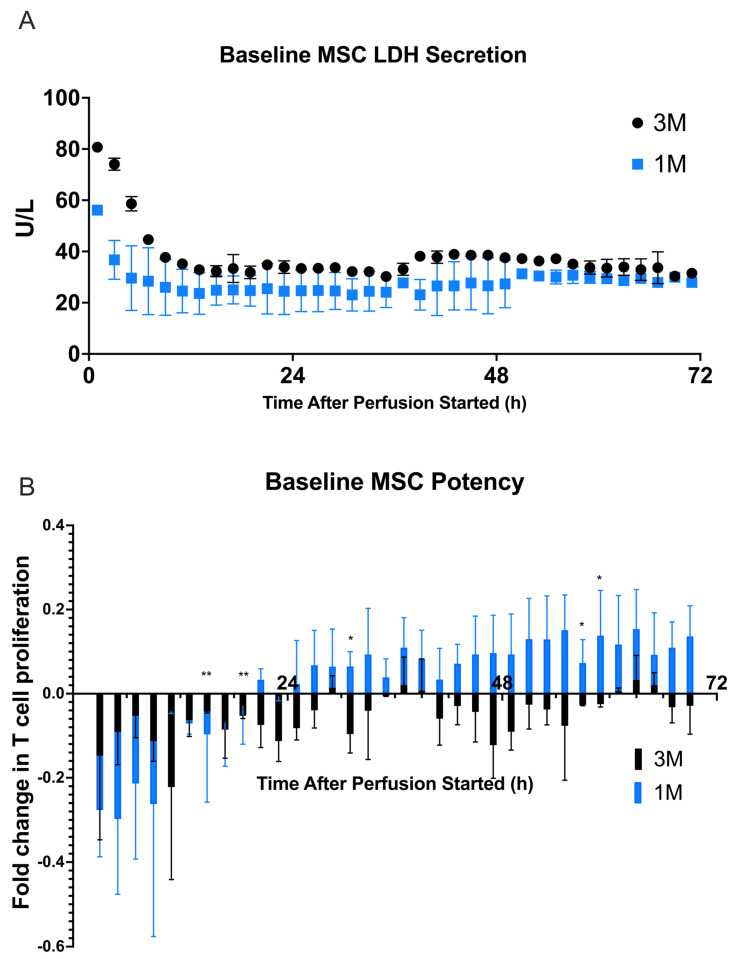
Baseline dynamic LDH secretion and potency of MSC secretome from bioreactors seeded with 1 × 10^6^ (▪) and 3 × 10^6^ (●) viable MSCs. (**A**) Baseline LDH (U/L) secretion by MSCs during 72 h continuous perfusion was quantified from each fraction by Cedex Bio Analyzer. (**B**) Fractions were cocultured with CD3/CD28–stimulated and CFSE–stained PBMCs for 72 h before being analyzed via flow cytometry for T cell proliferation. Fold change in T cell proliferation for each coculture compared to a negative control dose of 0 M was quantified in FlowJo. The figure shows the dynamics of T cell proliferation per fraction coculture with standard deviations for each. Statistical significance indicated by: * = *p*-value < 0.05 and ** = *p*-value < 0.01. A mixed-effects ANOVA was conducted to find statistically significant differences between the three cell dosages (0 M, 1 M, 3 M) and between the fractions of these dosages to determine which fractions were significantly different. *p*-values and results can be seen in Table 1.

**Figure 3 ijms-24-09602-f003:**
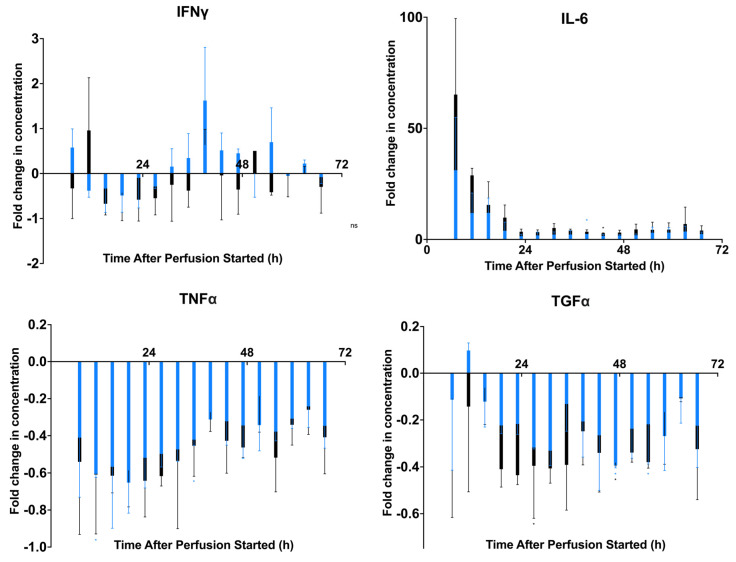
Dynamic secreted factor analysis from baseline MSC secretome fraction and PBMC 72 h coculture. Fractions were cocultured with CD3/CD28–stimulated and CFSE–stained PBMCs for 72 h. Measurement of secreted factors on the supernatants of the samples was performed via Milliplex Human Cytokine/Chemokine Magnetic Bead Panel. Fold change in concentrations compared to a negative control of IL-6 (pg/mL), IFNγ (pg/mL), TNFα (pg/mL), and TGFα (pg/mL) were quantified. Statistical significance indicated by: * = *p*-value < 0.05 and ** = *p*-value < 0.01. A table of *p*-values can be found in Appendix A.

**Figure 4 ijms-24-09602-f004:**
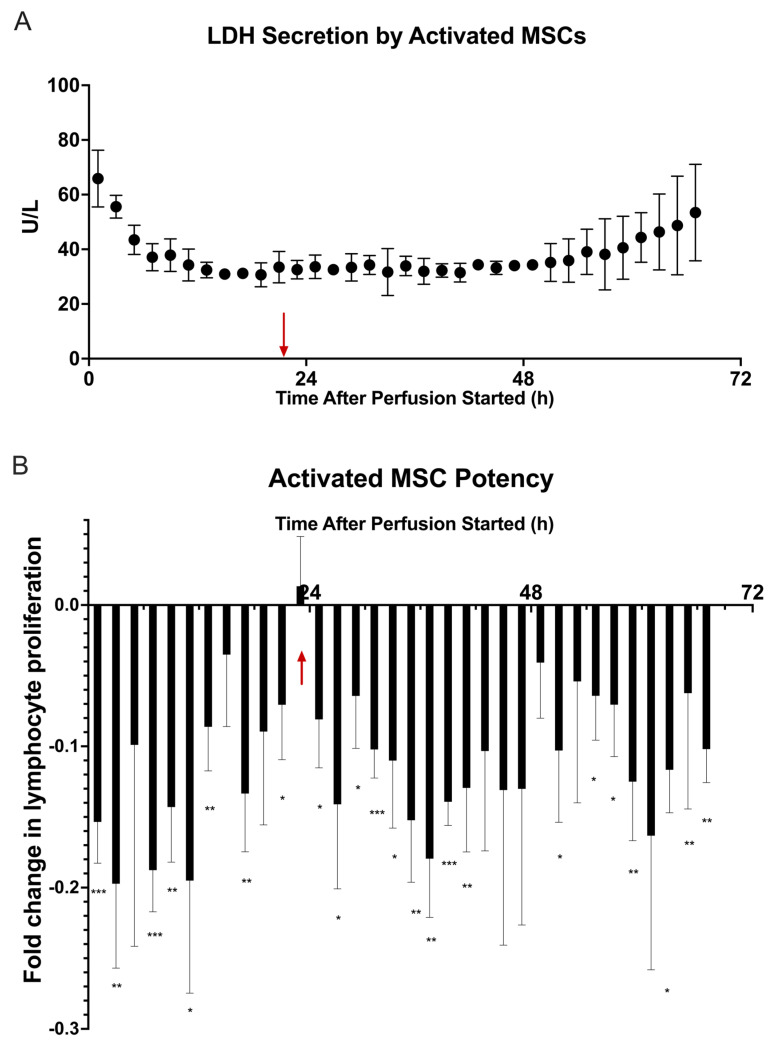
Dynamic LDH secretion and potency of MSC secretome with introduction of inflammatory medium at 24 h of perfusion. Bioreactors were seeded with 0 M and 3 × 10^6^ (**●**) viable MSCs, incubated for 2 h at 37 °C, and then perfused with exosome–depleted α-MEM for 24 h. At 24 h of perfusion, exosome–depleted α-MEM containing 10 ng/mL each of TNFα, IFNγ, and IL-1β was introduced to the system and perfused for the remaining 48 h of perfusion. Fractions of the MSC secretome were collected every 2 h. The fraction timepoint of the introduction of the inflammatory medium is indicated by a red arrow (

). (**A**) LDH (U/L) secretion was quantified from each fraction by Cedex Bio Analyzer. (**B**) Fractions were cocultured with CD3/CD28–stimulated and CFSE–stained PBMCs for 72 h before being analyzed via flow cytometry for T cell proliferation. Fold change in T cell proliferation for each 3 M cell dose coculture compared to a negative control dose of 0 M was quantified in FlowJo. The figure shows the dynamics of T cell proliferation per fraction coculture with standard deviations for each. A multiple unpaired student’s *t*-test was conducted on the two cell dosage groups for all timepoints, showing a significant difference in cell dosage potency between 0 M and 3 M cell groups. Statistical significance indicated by: * = *p*-value < 0.05, ** = *p*-value < 0.01, *** = *p*-value < 0.001. A table of *p*-values can be found in Appendix A.

**Figure 5 ijms-24-09602-f005:**
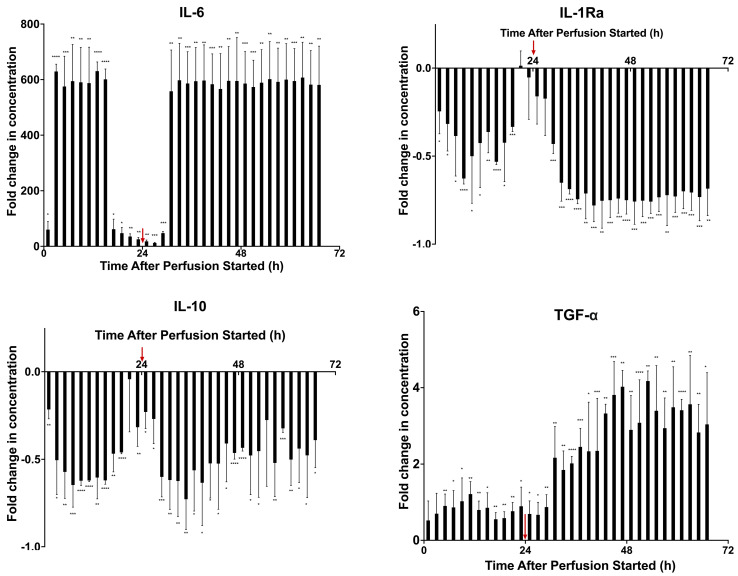
Dynamic secreted factor analysis from MSC secretome fraction and PBMC from inflamed model 72 h coculture. Fractions were cocultured with CD3/CD28–stimulated and CFSE–stained PBMCs for 72 h. Measurement of secreted factors on the supernatants of the samples was performed via Milliplex Human Cytokine/Chemokine Magnetic Bead Panel. Fold change in concentrations compared to a negative control of IL-6 (pg/mL), IL-1Ra (pg/mL), IL-10 (pg/mL), and TGFα (pg/mL) were quantified. Statistical significance indicated by: * = *p*-value < 0.05, ** = *p*-value < 0.01, *** = *p*-value < 0.001, **** = *p*-value < 0.0001. A table of *p*-values can be found in Appendix A.

**Figure 6 ijms-24-09602-f006:**
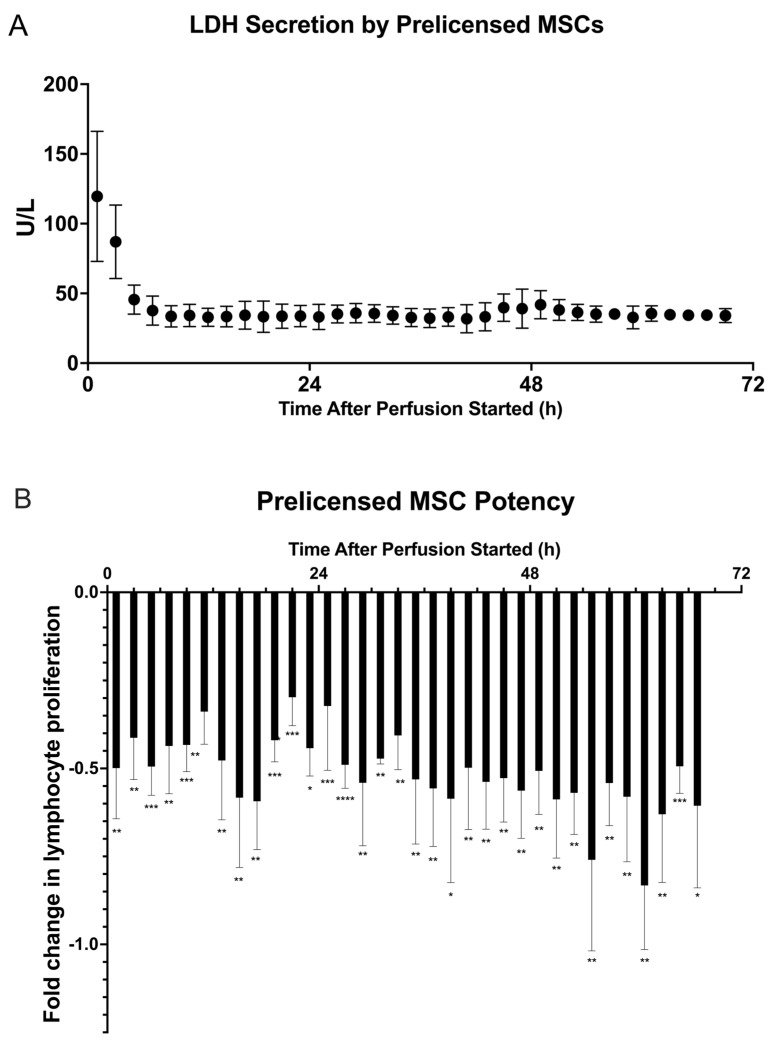
Dynamic LDH secretion and potency of prelicensed MSC secretome during 72 h bioreactor continuous perfusion. MSCs were cultured in inflammatory medium containing 10 ng/mL each of TNFα, IFNγ, and IL-1β for 24 h. Cells were trypsinized and bioreactors were seeded with 0 M and 3 × 10^6^ (●) viable MSCs, incubated for 2 h at 37 °C, and then perfused with exosome–depleted α-MEM for 72 h. Fractions of the MSC secretome were collected every 2 h. (**A**) LDH (U/L) secretion was quantified from each fraction by Cedex Bio Analyzer. (**B**) Fractions were cocultured with CD3/CD28–stimulated and CFSE–stained PBMCs for 72 h before being analyzed via flow cytometry for T cell proliferation. Fold change in T cell proliferation for each prelicensed 3 M cell dose coculture compared to a negative control dose of 0 M was quantified in FlowJo. The figure shows the dynamics of T cell proliferation per fraction coculture with standard deviations for each. A multiple unpaired student’s *t*-test was conducted on the two cell dosage groups for all timepoints, showing a significant difference on cell dosage potency between the 0 M and 3 M cell groups. Statistical significance indicated by: * = *p*-value < 0.05, ** = *p*-value < 0.01, *** = *p*-value < 0.001, **** = *p*-value < 0.0001. A table of *p*-values can be found in Appendix A.

**Figure 7 ijms-24-09602-f007:**
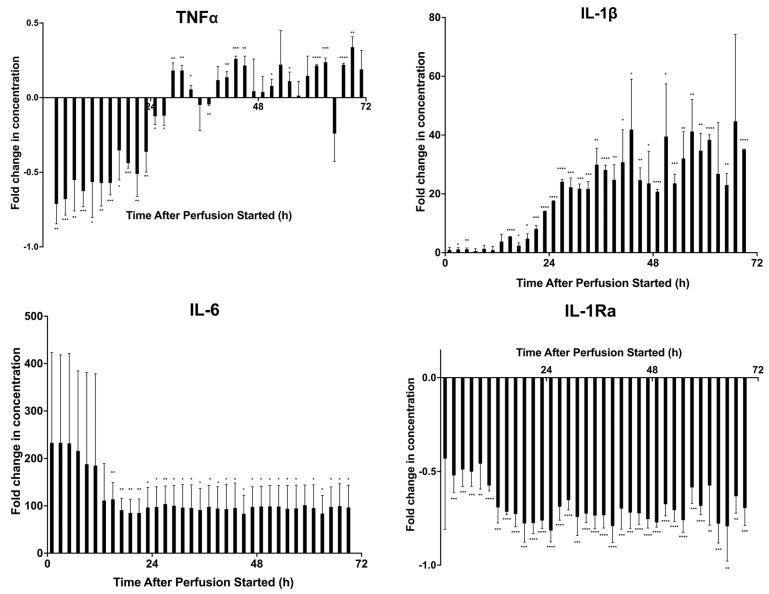
Dynamic secreted factor analysis from prelicensed MSC secretome fraction and PBMC 72 h coculture. Fractions were cocultured with CD3/CD28–stimulated and CFSE–stained PBMCs for 72 h. Measurement of secreted factors on the supernatants of the samples was performed via Milliplex Human Cytokine/Chemokine Magnetic Bead Panel. Fold change in concentrations compared to a negative control of TNFα (pg/mL), IL-1β (pg/mL), IL-6 (pg/mL), and IL-1Ra (pg/mL) were quantified. Statistical significance indicated by: * = *p*-value < 0.05, ** = *p*-value < 0.01, *** = *p*-value < 0.001, **** = *p*-value < 0.0001. A table of *p*-values can be found in Appendix A.

**Figure 8 ijms-24-09602-f008:**
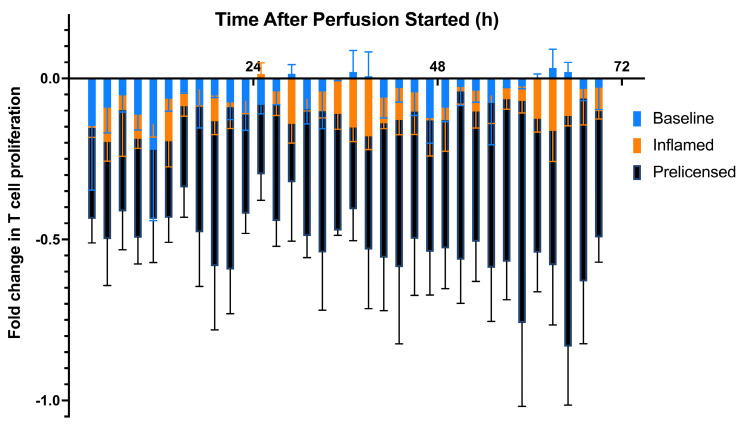
Comprehensive dynamic potency of 3 M MSC cell dosage from each perfusion model (Baseline; Inflamed via introduction of 10 ng/mL of IFNγ, 10 ng/mL of TNFα, and 10 ng/mL of IL-1β after 24 h of perfusion; Prelicensed with 10 ng/mL of IFNγ, 10 ng/mL of TNFα, and 10 ng/mL of IL-1β before perfusion). Fold changes in T cell proliferation (3 M vs. 0 M cell dosages) from each potency assay are plotted on one figure for comparison between the three potency studies conducted.

**Table 1 ijms-24-09602-t001:** One-way repeated measures ANOVA with Tukey’s multiple comparisons was conducted between fraction groups at different time points to determine how the impact of cell dose changed each day of perfusion.

	0–24 h	25–48 h	49–72 h
0 M vs. 1 M	*p* = 0.0170	*p* < 0.0001	*p* < 0.0001
0 M vs. 3 M	*p* = 0.0003	*p* < 0.0266	*p* = 0.0932
1 M vs. 3 M	*p* = 0.7862	*p* < 0.0001	*p* < 0.0001

## Data Availability

Data is contained within the article or Appendix A.

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
