# Peer review of "Dynamics of Ex Vivo Mesenchymal Stromal Cell Potency under Continuous Perfusion"

_ijms, 2023, doi:10.3390/ijms24119602_

Round 1
Reviewer 1 Report
Dynamics of ex vivo mesenchymal stromal cell potency under continuous perfusion
Thank you for the opportunity to review this manuscript detailing use of a continuous perfusion system to study temporal dynamics of MSC viability and immunomodulatory potency of MSC conditioned media in a lymphocyte proliferation assay. Comments generally relate to clarity of study design and presentation of findings, specifically justification of cytokines and pre-licensing agents selected. See specific comments below.
15 – would recommend to indicate more specifically (briefly) here what is indicated as potent (i.e. what outcomes were greater in the first 24 hours) and what pre-licensing agents were used to achieve that – it reads somewhat vague currently although it did pique my interest to continue reading into the article
41 – remove s before seeded
88 – was minimum viability level maintained or did it just not change significantly
93 – how was this flow rate determined (0.5ml/h)?
Tables and Figures – recommend to simply list P<0.05 as significant * and then state what the p-values are rather than levels of significance which is irrelevant (values are either significant or not)
Throughout revise gamma and alpha symbols to symbols rather than letters
327 – remove one of the ‘from the’
Figure 8 (and throughout) – what licensing agent was used? This is unclear at this point in the manuscript. As there are many that may be integrated and have been evaluated, this needs to be clarified
Please include greater discussion and justification of pre-licensing cocktail and add clarification earlier in the manuscript what the ‘licensed’ MSC were treated with – currently have to read until the methods until becomes clear that a single licensing cocktail was used, this needs to be clarified and limitation addressed that each was not assessed separately or present preliminary data where this has already been done
Further discussion of which cytokines were selected is indicated – Have to read until methods section to see that 10 cytokines were assessed – However, others would perhaps have been more interesting in this clinical context (i.e. IL8, MCP1) towards improved immunomodulation and recruitment of neutrophils and macrophages etc. Please expand on the selection of those investigated here.
Expand on significance of the significance of LDH release profile and why this was included as an outcome measure versus other potential indications of function
Recommend to present statistical comparisons (rather than just relative number of stars in figures) either in results section or table
Recommend adding a study overview figure to clarify different portions of study design – this could be potentially added to current figure 1 of the bioreactor
Reviewer 2 Report
This is an interesting and well-prepared studies overall. In this study, the researchers investigated the temporal variations in the effectiveness of MSC secretome within an ex vivo hollow fiber bioreactor employing a continuous perfusion cell culture technique. The objective was to assess the changes in MSC secretome potency over time under various conditions, such as baseline, activation with a pro-inflammatory stimulus, and pre-licensing.
The findings reveal that MSC secretome is most effective during the initial 24 hours, and this effectiveness is further enhanced when MSCs undergo pre-licensing. And, the study demonstrated that pre-licensed MSCs exhibit the highest lymphocyte suppression among the three perfusion conditions analyzed.
The reviewer observes that this manuscript appears to be a continuation of the research group's previous work, as the experimental setup is distinct. Providing additional information about the experimental design would enhance the clarity of the paper, particularly for readers who do not have access to the Journal of Visualized Experiments (JOVE).
It would be beneficial to comment on the potential translation of these findings to in vivo settings. Furthermore, a brief discussion of the possible impact of the results on clinical applications would be valuable.
The phrase "we can see" should be avoided in favor of a more scholarly tone.
Reviewer 3 Report
The presented paper describes the dynamics of mesenchymal stromal cells (MSC) secretome potency in an ex vivo hollow fiber bioreactor using a continuous perfusion cell culture system that fractionated MSC secreted factors over time. The article appears to be based on Aneesha Doshi's master of science' thesis (2022) entitled 'Dynamics of Mesenchymal Stromal Cell Potency During Ex Vivo Bioreactor Therapy'. The article is well written and consistent with the development of the modulation of immune responses in vitro.
Some minor comments: 1. Lines 113, 125: The abbreviation '0M' is misleading. Additional clarification is needed that these are fibers without cells. 2. Line 433. The product code (catalog number) of MSCs should be included in the paragraph. 3. Lines 439-442: The data presented are derived from the cell product description? 4. Line 522. Do the authors have data on the phenotype of the immune cells populations? 5. Line 567. The authors used unconventional statements such as " ** = significance at 0.01 level". I recommend using the p-value description, e.g. 'p-value<0.01'. 6. Figure 2b. The authors need to explain in the figure caption what signatures 1A, 2A and 3A mean. 7. Figure 2b. The abbreviation 'μl' should be used instead of 'ul'.Some major comments: I recommend including more 2019-2023 sources in the introduction and discussion for the next topics: 1. Paracrine effects of MSC therapy not only affect the reduction of inflammation, but also epithelialization and vascularization in affected tissues [https://doi.org/10.3390/dj9090101] [https://doi.org/10.3390/cells10071729]
2. Can the proposed approaches explain the pathogenesis of adverse events, side effects and complications of MSC-based and secretome-based therapies? [https://doi.org/10.1186/s13287-021-02609-x] [https://doi.org/10.21037/sci-2022-025] 3. I find it odd that the authors' previous paper with the same materials and methods on this topic was not properly discussed [Allen, A., Vaninov, N., Li, M. et al. Mesenchymal Stromal Cell Bioreactor for Ex Vivo Reprogramming of Human Immune Cells. Sci Rep 10, 10142 (2020). https://doi.org/10.1038/s41598-020-67039-w]
